# Association between early viral LRTI and subsequent wheezing development, a meta-analysis and sensitivity analyses for studies comparable for confounding factors

Sebastien Kenmoe[1]*, Arnol Bowo-Ngandji[2], Cyprien Kengne-Nde[3], Jean Thierry Ebogo-Belobo[4], Donatien Serge Mbaga[2], Gadji Mahamat[2], Cynthia Paola Demeni Emoh[2], Richard Njouom[1]*

1 Department of Virology, Centre Pasteur of Cameroon, Yaoundé, Cameroon, 2 Faculty of Science, Department of Microbiology, The University of Yaounde I, Yaoundé, Cameroon, 3 National AIDS Control Committee, Epidemiological Surveillance, Evaluation and Research Unit, Yaounde, Cameroon, 4 Medical Research Centre, Institut of Medical Research and Medicinal Plants Studies, Yaoundé, Cameroon

* kenmoe@pasteur-yaounde.org (SK); njouom@pasteur-yaounde.org (RN)

## Abstract

### Introduction

Consideration of confounding factors about the association between Lower Respiratory Tract Infections (LRTI) in childhood and the development of subsequent wheezing has been incompletely described. We determined the association between viral LRTI at $\leq$ 5 years of age and the development of wheezing in adolescence or adulthood by a meta-analysis and a sensitivity analysis including comparable studies for major confounding factors.

### Methods

We performed searches through Pubmed and Global Index Medicus databases. We selected cohort studies comparing the frequency of subsequent wheezing in children with and without LRTI in childhood regardless of the associated virus. We extracted the publication data, clinical and socio-demographic characteristics of the children, and confounding factors. We analyzed data using random effect model.

### Results

The meta-analysis included 18 publications (22 studies) that met the inclusion criteria. These studies showed that viral LRTI in children $\leq$ 3 years was associated with an increased risk of subsequent development of wheezing (OR = 3.1, 95% CI = 2.4–3.9). The risk of developing subsequent wheezing was conserved when considering studies with comparable groups for socio-demographic and clinical confounders.

### Conclusions

When considering studies with comparable groups for most confounding factors, our results provided strong evidence for the association between neonatal viral LRTI and the

**Data Availability Statement:** All relevant data are within the manuscript and its Supporting Information files.

**Funding:** The authors received no specific funding for this work.

**Competing interests:** The authors have declared that no competing interests exist.

subsequent wheezing development. Further studies, particularly from lower-middle income countries, are needed to investigate the role of non-bronchiolitis and non-HRSV LRTI in the association between viral LRTI in childhood and the wheezing development later. In addition, more studies are needed to investigate the causal effect between childhood viral LRTI and the wheezing development later.

## Trial registration

Review registration: PROSPERO, CRD42018116955; https://www.crd.york.ac.uk/prospero/display_record.php?ID=CRD42018116955.

## Introduction

Epidemiological studies have shown that up to 50% of children have at least one episode of wheezing before their third birthday [1–3]. This wheezing is responsible for hospitalization rates in children of up to 8% [1].

Several studies have shown that hospitalization for early Human Respiratory Syncytial Virus (HRSV) bronchiolitis was a predictor of subsequent wheezing episodes [4,5]. The association between HRSV-bronchiolitis and long-term wheezing is now clearly established by multiple original studies and meta-analyses [4,5]. The current residual questions are to known whether the HRSV-LRTI are for subsequent wheezing: 1) a marker of susceptibility, 2) a causal agent, or 3) both a marker of susceptibility and a causal agent. The responses to these residual shadow points now relate to the multiple confounding factors or ideally the response to the causal effect between the HRSV-bronchiolitis and wheezing which can only be addressed by interventional studies with prophylactic means such as palivizumab or very soon with anti-HRSV vaccines approval [6]. A recent meta-analysis failed to provide acceptable evidence for the causal relationship between childhood HRSV infections and long-term respiratory morbidity [7]. This question of wheezing after bronchiolitis in childhood have also been incompletely explored in multiple aspects including the age of children at the time of bronchiolitis development, bronchiolitis due to non-HRSV viruses, non-bronchiolitis Lower Respiratory Tract Infections (LRTI), and the influence of confounding factors [8–10]. The introduction of molecular diagnostic tests has led to better definition of the role of respiratory virus in bronchiolitis [11]. It is now recognized that Human Metapneumovirus (HMPV), Human Bocavirus (HBoV) or even Rhinovirus (RV) are among common viruses involved in bronchiolitis [12–16]. It has also been shown that childhood RV infections are associated with subsequent wheezing [17,18]. Apart from HRSV and RV, there is no systematic review with respect to the association between childhood LRTI due to other common respiratory virus and later wheeze. In addition to bronchiolitis, other low respiratory infections are very common in childhood [19,20]. To date, however, there is no systematic review on the influence of non-bronchiolitis respiratory infections in infancy on long-term wheezing. There are several confounding factors that have not yet been explored concerning their effect on post bronchiolitis wheezing [21–25]. These confounding factors include atopic predisposition [26–34], reduced premorbid lung function [35], smoking exposure, breastfeeding [34,36], premature birth, low birth weight, or overcrowding [36,37]. These confounding factors may act synergistically with low respiratory infections in childhood and all these mechanisms remain to be elucidated [38].

Early recognition of children at risk of developing long-term wheezing may be important for many prevention and management policies [39,40]. The aim of this study was to evaluate,

in a meta-analysis and sensitivity analyses including studies comparable for sociodemographic and clinical confounding factors, wheezing as a long-term sequela of viral LRTI in children.

## Methods

### Study design

This systematic review was prepared according to the Centre for Reviews and Dissemination guidelines [41]. The PRISMA declaration served as a model for the presentation of this review (S1 Table) [42]. The protocol of this review has been registered in the PROSPERO database under number CRD42018116955. Obtaining ethical clearance was not required for this study.

### Inclusion and exclusion criteria

Due to the multiple limitations of cross-sectional (non-comparable groups) and case control (misclassification and recall bias) studies in estimating the association between exposure and outcome, we include only cohort studies that compare children with LRTI with controls in regards of wheezing as long-term respiratory sequelae.

   We considered studies with healthy controls (with no history of LRTI or non-respiratory conditions) compared to LRTI cases with criteria as similar as possible (gender, age, geographic origin, geographic location, social class and time of inclusion). The study participants were children with severe respiratory infections and independently of the respiratory virus detected. The definitions of LRTI were adapted as described by the authors of the primary studies.We excluded studies with high-risk people (preterm, chronic respiratory disease, heart disease or immunodeficiency) to reduce the potentially confounding effect of co-morbidities in these individuals on the association between LRTI and wheezing.

### Study outcomes and case definition

The main outcome was the link between LRTI in children at $\leq 5$ years and the wheezing development later. The secondary results were to determine factors associated with the development of wheezing after LRTI in infancy. Wheezing was considered as any whistling nose at the expiration declared by the included study. Wheezing during the interview was defined as current wheezing. Wheezing in the last 12 months was defined as wheezing episodes during the previous year. Recurrent wheezing was considered for persistent wheezing between the diagnosis of LRTI and the interview. Recurrent wheezing was also considered as $\geq 3$ episodes of bronchial obstruction. If there were multiple wheezing phenotypes in the same study, we chose the one that spanned the longest time. We separately considered studies with multiple effect data depending on interview time, respiratory viruses detected or types of LRTI.

### Search strategy

Research was conducted in the following databases: Pubmed and Global Index Medicus. The search strategy in Pubmed is shown in S2 Table. This search strategy was adapted for the second database. The search was performed without any language restrictions since the creation of the databases until the 28 August 2020. The reference lists selected and the relevant review was deeply checked to include additional articles.

### Study selection

Two investigators (JTEB and SK) independently selected potentially relevant studies based on the titles and abstracts from the list of references in Rayyan website [43]. Full versions of

selected articles were uploaded. The selection process was summarized in a PRISMA flowchart.

### Data extraction

Six authors (SK, ABN, JTEB, DSM, GM, and CPDE) evaluated independently all included studies. The following information was collected: 1) title, first author, year of publication, time of data collection, country and participants interview period of article; 2) the type, rank, period, age, hospitalization and type of infection associated with the LRTI; 3) the age and gender of controls; 4) the total number of cases and controls and numbers with wheezing or confounders at follow-up. All data on confounding factors collected were defined as presented in the included article. We have harmonized the names of the confounding factors collected in several articles according to their similarity.

### Appraisal of the methodological quality of included studies and risk of bias

Six authors (SK, ABN, JTEB, DSM, GM, and CPDE) independently assessed the quality of each study using an adapted version of Newcastle-Ottawa Scale (S3 Table) [44]. Discrepancies in the study selection, study quality evaluation, and data extraction were resolved by discussion between the authors involved in the process or by consultation of another author of the review as appropriate (SK).

### Data synthesis and analysis

The odds ratio (OR) and the 95% confidence interval (95% CI) and prediction interval were used as a measure of the association between LRTI and long-term wheezing. These parameters were calculated using a random-effects model by the Der Simonian and Laird method [45,46]. Heterogeneity was assessed by the Q test p-value and the $I^2$ statistic [47,48]. The heterogeneity between the studies was considered significant for the values of $P < 0.1$ and $I^2 > 50\%$. The robustness of the results was assessed by a sensitivity analysis including only the first episode of LRTI, physician-diagnosed wheezing and studies with proportions of confounding factors comparable between cases and controls. The comparability of confounders between cases and controls were estimated using the Chi-square and Fisher tests for qualitative variables and Student's T-test for continuous variables. The primary confounding factors data for continuous variables expressed as median and/or range were converted to mean and standard deviation [49]. Univariate subgroup analyses were performed using random effect meta-analyses based on sampling method, time of exposure collection, age range during LRTI, age at time of interview for wheezing, hospitalization of controls, viruses responsible for LRTI, type of LRTI, and type of wheezing [45]. We analyzed the data using the "meta: version 4.15–0" and "metafor: version 2.4–0" packages in version 3.5.1 of software R [50,51].

## Results

### Study selection

The search strategy identified 1962 articles. We eliminated 1425 articles unrelated to the objectives of the study. We reviewed a total of 139 complete articles to determine their eligibility for meta-analysis. Among these items, we excluded 121 for multiple reasons (S4 Table) and included 18 in this study (Fig 1) [52–69]. The 18 citations represented 22 individual studies since 3 references included data collected at 2 different ages and one study included data for 2 different viruses [54,55,61,68].

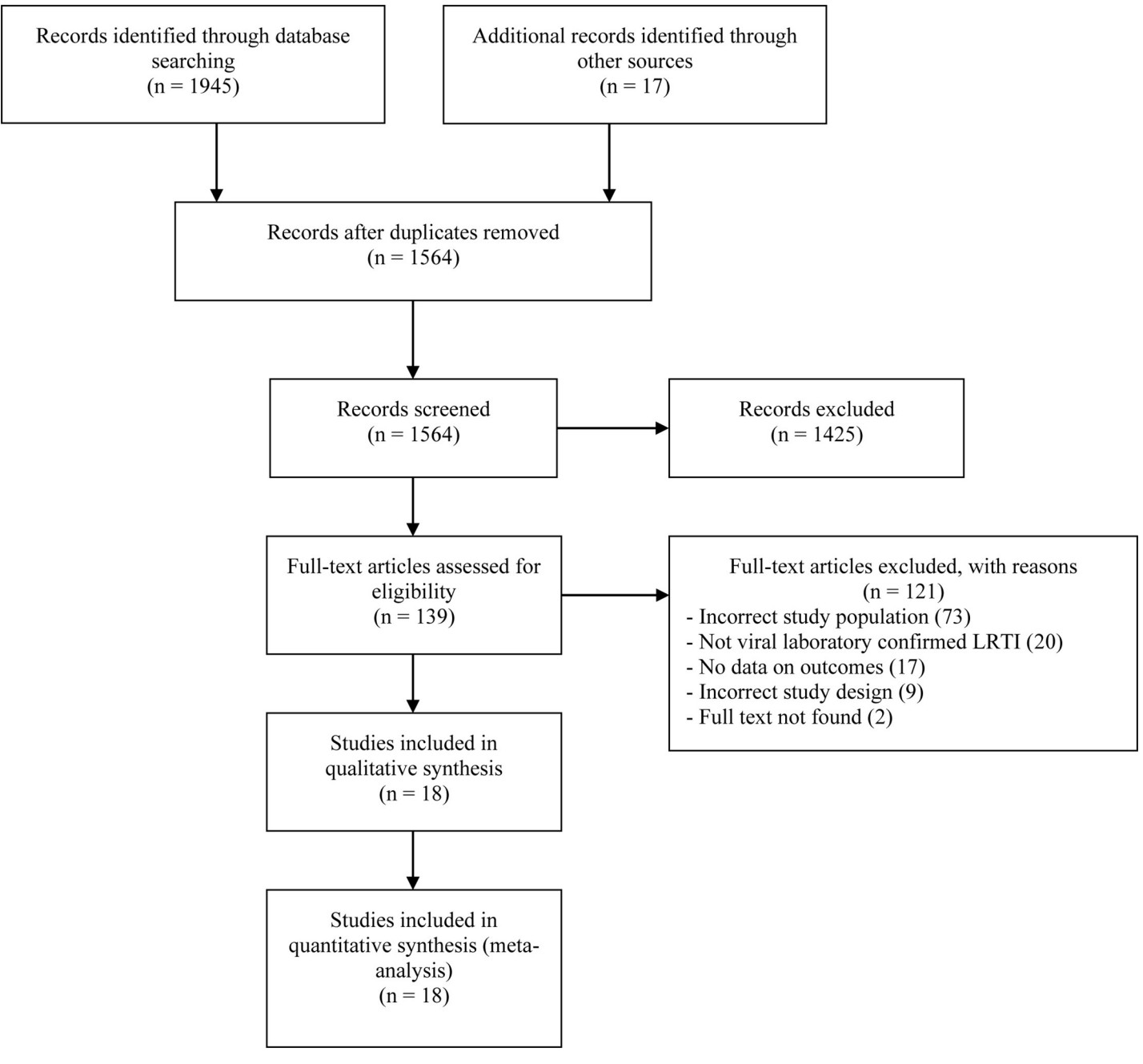

**Fig 1. Flow chart for the systematic literature search.**

## Characteristics of studies included for the meta-analysis

S5 Table summarizes the characteristics of the 22 included studies. The included cohort studies were published between 1978 and 2017 and most were conducted in Northern Europe (68.2%) and in high income countries (95.5). Most of the included studies had prospective (77.3%), non-probabilistic (95.5%) recruitment and follow-up duration ranged from 1.5 to 10 years. All children with LRTI were hospitalized, had LRTI from 1960 to 2005, and most were < 1 year (59.1%) with their first episode (68.2%) of bronchiolitis (68.2%) due to HRSV (90.9%). Wheezing in children was reported between 2 and 20 years and most studies were between 2 and 10

years (77.2%). The most reported wheezing phenotype in the included studies was recurrent wheezing (59.1%). All included studies were at low risk of bias (S6 Table). S7 Table depicts individual data of included studies.

## Meta-analysis results

**Association between LRTI in infancy and wheezing later.** Findings from the meta-analysis showed that the rate of wheezing was significantly higher in patients with childhood LRTI than in controls: OR = 3.0, 95% CI = 2.3–3.9 (Fig 2). All wheezing categories (any wheezing:

| Study or Subgroup | LRTI + Events | Total | LRTI − Events | Total | Weight | OR [95% CI] |
|---|---|---|---|---|---|---|
| **Any wheezing** | | | | | | |
| Sigurs, 1995, 1 year | 19 | 47 | 15 | 93 | 5.4% | 3.53 [1.58; 7.88] |
| Sigurs, 1995, 3 years | 28 | 47 | 30 | 93 | 6.0% | 3.09 [1.50; 6.40] |
| **Random effect meta-analysis** | | 94 | | 186 | 11.4% | 3.28 [1.92; 5.63] |
| Heterogeneity: Tau$^2$ = 0; Chi$^2$ = 0.06, df = 1 (P = 0.8123); I$^2$ = 0% | | | | | | |
| **Current wheezing** | | | | | | |
| Henderson, 2005, 30–42 months | 13 | 95 | 305 | 9786 | 7.2% | 4.93 [2.72; 8.94] |
| Henderson, 2005, 69–81 months | 1 | 62 | 153 | 7189 | 1.4% | 0.75 [0.10; 5.47] |
| Sims, 1978 | 18 | 35 | 1 | 35 | 1.2% | 36.00 [4.43; 292.85] |
| Singleton, 2003 | 41 | 95 | 41 | 113 | 7.6% | 1.33 [0.76; 2.33] |
| Stensballe, 2017, 1,5 years | 33 | 39 | 13 | 23 | 3.2% | 4.23 [1.28; 14.02] |
| Stensballe, 2017, 5 years | 29 | 33 | 13 | 18 | 2.3% | 2.79 [0.64; 12.11] |
| **Random effect meta-analysis** | | 359 | | 17164 | 23.0% | 3.12 [1.56; 6.26] |
| Heterogeneity: Tau$^2$ = 0.3868; Chi$^2$ = 18.04, df = 5 (P = 0.0029); I$^2$ = 72.3% [36.0%; 88.0%] | | | | | | |
| **Recurrent wheezing** | | | | | | |
| Bertrand, 2015 | 4 | 14 | 2 | 5 | 1.2% | 0.60 [0.07; 5.06] |
| Fjaerli, 2005 | 17 | 35 | 9 | 64 | 4.3% | 5.77 [2.19; 15.19] |
| García–García, 2007, HMPV | 20 | 23 | 12 | 30 | 2.4% | 10.00 [2.43; 41.22] |
| García–García, 2007, HRSV | 24 | 32 | 12 | 30 | 3.7% | 4.50 [1.52; 13.30] |
| Juntti, 2003 | 47 | 76 | 36 | 76 | 6.7% | 1.80 [0.94; 3.43] |
| Kristjánsson, 2006 | 11 | 17 | 12 | 25 | 2.9% | 1.99 [0.56; 7.05] |
| Osundwa, 1993 | 31 | 70 | 9 | 70 | 5.1% | 5.39 [2.32; 12.53] |
| Poorisrisak, 2010 | 44 | 74 | 39 | 74 | 6.7% | 1.32 [0.69; 2.52] |
| Pullan, 1982 | 55 | 130 | 21 | 111 | 7.3% | 3.14 [1.74; 5.66] |
| Sigurs, 2000 | 13 | 47 | 10 | 93 | 4.6% | 3.17 [1.27; 7.93] |
| Sigurs, 2005 | 14 | 47 | 15 | 93 | 5.1% | 2.21 [0.96; 5.08] |
| Sigurs, 2010 | 18 | 46 | 8 | 92 | 4.5% | 6.75 [2.65; 17.21] |
| Sly, 1984 | 13 | 20 | 8 | 20 | 2.9% | 2.79 [0.77; 10.04] |
| **Random effect meta-analysis** | | 631 | | 783 | 57.4% | 3.03 [2.19; 4.20] |
| Heterogeneity: Tau$^2$ = 0.1272; Chi$^2$ = 21.42, df = 12 (P = 0.0445); I$^2$ = 44% [ 0.0%; 70.7%] | | | | | | |
| **Wheezing in the last 12 months** | | | | | | |
| Zomer–Kooijker, 2014 | 33 | 155 | 42 | 516 | 8.3% | 3.05 [1.86; 5.02] |
| **Random effect meta-analysis** | | 155 | | 516 | 8.3% | 3.05 [1.86; 5.02] |
| Heterogeneity: not applicable | | | | | | |
| **Overall random effect meta-analysis** | | 1239 | | 18649 | 100.0% | 3.04 [2.37; 3.89] |
| Residual heterogeneity: Tau$^2$ = NA; Chi$^2$ = 39.52, df = 18 (P = 0.0024); I$^2$ = 54.5% [23.5%; 72.9%] | | | | | | |

0.01   0.1   1   10   100

**Fig 2. Forest plot of wheezing in children with and whitout LRTI in infancy.**

OR = 3.2, 95% CI = 1.9–5.3, $I^2$ = 0.0%; current wheezing: OR = 3.1, 95% CI = 1.5–6.2, $I^2$ = 72.3%; recurrent wheezing: OR = 3.0, 95% CI = 2.2–4.2, $I^2$ = 44.0%; wheezing in the last 12 months: OR = 3.0, 95% CI = 1.8–5.0 were significantly more common in older LRTI patients compared to the control group.

### Sensitivity analysis

We assessed the constancy of our results through multiple sensitivity analyses including studies reporting only the first episode of LRTI, physician-diagnosed wheezing, and studies comparable between cases and controls for multiples confounding factors (S8 and S9 Tables). Table 1 presents the results of these different sensitivity analyses. The results of the sensitivity analyses were consistent with those of the overall findings for studies reporting only the first episode of LRTI and studies with studies comparable between cases and controls for almost all confounding factors. We found no publication bias in the overall analyses (p-value Egger test = 0.282). Funnel graph showed no asymmetry (S1 Fig). Significant heterogeneity was observed in overall analyses (I2: 47.4% 95% = 13.5–68; p-value heterogeneity: 0.008).

### Subgroup analysis of the occurrence of wheezing in children with and without LRTI in infancy

The type of LRTI was associated with the occurrence of wheezing (p = 0.007) (S10 Table). History of bronchiolitis (OR = 3.8, 95% CI = 3.4–4.7) and LRTI unspecified (OR = 1.9, 95% CI = 1.4–2.7) were associated with the occurrence of wheezing in childhood. Age rage at recruitment was significantly associated with the occurrence of wheezing later (p <0.045). We did not record any difference in the occurrence of wheezing by the children age range at recruitment (p = 0.113) and the type of virus screened (p = 0.249). However, the only study available with Adenovirus type 7 was not associated with the occurrence of wheezing according to the history of LRTI (OR = 2.8, IC 95% = 0.8–10.0). No difference was recorded in the occurrence of wheezing depending on the age range at the interview (p = 0.053), the sampling approach (non-probabilistic vs probabilistic, p = 0.505), the time of exposure collection (prospective vs retrospective, p = 0.990), the hospitalization of controls (p = 0.478), and the type of wheezing (p = 0.996). However, all age groups at interview up to 20 years were associated with an increased risk of developing wheezing in former cases of LRTI compared to controls.

## Discussion

This systematic review of studies with LRTI patients in nearly half a century gives four main results. Children who experienced an episode of viral LRTI at ≤ 3 years of age were 3 times more likely than controls to develop wheezing later. This increased risk of wheezing in children with a history of viral LRTI was not influenced by the rank of LRTI episode, physician-diagnosed wheezing and studies comparable for the investigated confounding factors.

These results are consistent with previous systematic reviews that had already demonstrated an increased risk of wheezing among old bronchiolitis patients [8–10]. Kneyber et al have demonstrated in a quantitative analysis that hospitalized children with bronchiolitis episodes due to HRSV at <1 year of age had an increased risk of wheezing compared to controls [70]. Fauroux et al in a systematic review without meta-analysis of data published between 1995 and 2015 in Western countries reported that children hospitalized for HRSV infections were at high risk of developing recurrent wheezing until age 25 years old [71]. Shi et al in a recent meta-analysis showed that childhood HRSV infection was significantly associated with wheezing later in former LRTI cases compared to healthy individuals or controls without respiratory infection in infancy [8].

**Table 1. Wheezing in children with LRTI in infancy and control without respiratory diseases.**

| Wheezing | OR (95%CI) | 95% Prediction interval | N Studies | N LRTI cases | N controls | H (95%CI) | $I^2$ (95%CI) | P heterogeneity | P Egger test |
|---|---|---|---|---|---|---|---|---|---|
| Overall | 3 [2.4–3.9] | [1.4–6.7] | 22 | 1239 | 18649 | 1.4 [1.1–1.8] | 47.4 [13.5–68] | 0.008 | 0.282 |
| Sensitivity analyses | | | | | | | | | |
| First episode of LRTI | 2.8 [2.1–3.7] | [1.3–6.1] | 15 | 958 | 18316 | 1.4 [1–1.9] | 46.7 [2.8–70.8] | 0.024 | 0.582 |
| Physician-diagnosed wheezing | 3 [1.9–5] | [0.9–10.8] | 7 | 363 | 340 | 1.5 [1–2.3] | 55.4 [0–80.9] | 0.036 | 0.993 |
| Studies comparable for allergic rhinitis | 3.4 [1.6–7.4] | [0–7574.6] | 3 | 131 | 136 | 1.7 [1–3.1] | 64.5 [0–89.8] | 0.06 | 0.094 |
| Studies comparable for antiasthmatic treatment | 3.1 [1.7–5.7] | NA | 1 | 130 | 111 | NA | NA | 1 | NA |
| Studies comparable for asthma admissions | 4.5 [1.5–13.3] | NA | 1 | 32 | 30 | NA | NA | 1 | NA |
| Studies comparable for asthma in children | 1.3 [0.7–2.5] | NA | 1 | 74 | 74 | NA | NA | 1 | NA |
| Studies comparable for asthma in father | 6 [2.6–14.3] | NA | 2 | 55 | 60 | 1 | 0 | 0.38 | NA |
| Studies comparable for asthma in mother | 6 [2.6–14.3] | NA | 2 | 55 | 60 | 1 | 0 | 0.38 | NA |
| Studies comparable for asthma in parents | 3.4 [2.3–5.1] | [1.4–8.4] | 4 | 187 | 371 | 1 [1–2.6] | 5.2 [0–85.5] | 0.367 | 0.555 |
| Studies comparable for asthma in siblings | 6 [2.6–14.3] | NA | 2 | 55 | 60 | 1 | 0 | 0.38 | NA |
| Studies comparable for atopic dermatitis | 3.1 [2–4.8] | [1.1–8.6] | 7 | 295 | 361 | 1.3 [1–2] | 39.1 [0–74.4] | 0.131 | 0.131 |
| Studies comparable for atopy in children | 3 [1.4–6.4] | [0.2–53.4] | 4 | 149 | 154 | 1.7 [1–2.9] | 64.8 [0–88] | 0.037 | 0.168 |
| Studies comparable for atopy in father | 6 [2.6–14.3] | NA | 2 | 55 | 60 | 1 | 0 | 0.38 | NA |
| Studies comparable for atopy in mother | 3.6 [2.4–5.6] | [0.2–59.1] | 3 | 210 | 576 | 1.1 [1–3.5] | 22.6 [0–91.9] | 0.275 | 0.415 |
| Studies comparable for atopy in parents | 3.3 [2.3–4.7] | [2–5.3] | 7 | 273 | 417 | 1 [1–1.8] | 0 [0–70.1] | 0.438 | 0.357 |
| Studies comparable for atopy in siblings | 10 [2.4–41.2] | NA | 1 | 23 | 30 | NA | NA | 1 | NA |
| Studies comparable for current asthma | 5.8 [2.2–15.2] | NA | 1 | 35 | 64 | NA | NA | 1 | NA |
| Studies comparable for current atopy | 5.9 [3.1–11.3] | [0.1–382.6] | 3 | 90 | 124 | 1 [1–1.9] | 0 [0–73.2] | 0.679 | 0.794 |
| Studies comparable for current eczema | 5.8 [2.2–15.2] | NA | 1 | 35 | 64 | NA | NA | 1 | NA |
| Studies comparable for family history of atopy | 2 [0.6–7.1] | NA | 1 | 17 | 25 | NA | NA | 1 | NA |
| Studies comparable for heredity for asthma | 3.3 [2–5.2] | [0.2–66.1] | 3 | 141 | 279 | 1 [1–1] | 0 [0–0] | 0.97 | 0.605 |
| Studies comparable for heredity for atopy | 3.3 [2–5.2] | [0.2–66.1] | 3 | 141 | 279 | 1 [1–1] | 0 [0–0] | 0.97 | 0.605 |
| Studies comparable for history of asthma | 2.9 [1.3–6.2] | NA | 2 | 111 | 140 | 2 [1–4.1] | 74 [0–94.1] | 0.05 | NA |
| Studies comparable for history of atopy | 4.6 [2.8–7.5] | [0.2–110.8] | 3 | 192 | 17039 | 1.3 [1–2.4] | 42.2 [0–82.5] | 0.177 | 0.652 |
| Studies comparable for history of eczema | 2.9 [1.3–6.2] | NA | 2 | 111 | 140 | 2 [1–4.1] | 74 [0–94.1] | 0.05 | NA |

(*Continued*)

**Table 1.** (Continued)

| Wheezing | OR (95%CI) | 95% Prediction interval | N Studies | N LRTI cases | N controls | H (95%CI) | I² (95%CI) | P heterogeneity | P Egger test |
|---|---|---|---|---|---|---|---|---|---|
| Studies comparable for history of pertussis | 3.3 [1.9–5.6] | NA | 2 | 94 | 186 | 1 | 0 | 0.812 | NA |
| Studies comparable for male gender | 3.2 [2.4–4.3] | [1.4–6.9] | 15 | 768 | 1279 | 1.3 [1–1.8] | 45 [0–70] | 0.03 | 0.307 |
| Studies comparable for maternal smoking | 4.7 [2.5–8.9] | [1.2–19] | 4 | 127 | 101 | 1 [1–1.9] | 0 [0–71.5] | 0.657 | 0.466 |
| Studies comparable for maternal smoking during pregnancy | 1.8 [0.9–3.4] | NA | 1 | 76 | 76 | NA | NA | 1 | NA |
| Studies comparable for parental smoking | 5.8 [2.2–15.2] | NA | 1 | 35 | 64 | NA | NA | 1 | NA |
| Studies comparable for paternal smoking | 3.9 [2.4–6.3] | [0.2–90.4] | 3 | 185 | 171 | 1.1 [1–3.3] | 12.3 [0–90.9] | 0.32 | 0.247 |
| Studies comparable for pets at home | 3.1 [2.4–4.1] | [2.3–4.3] | 9 | 520 | 1116 | 1.1 [1–1.5] | 13 [0–54.9] | 0.326 | 0.18 |
| Studies comparable for premature birth | 3.9 [1.9–8] | [0–381.1] | 3 | 104 | 71 | 1 [1–1.2] | 0 [0–26.8] | 0.868 | 0.315 |
| Studies comparable for siblings in the house | 5.8 [2.2–15.2] | NA | 1 | 35 | 64 | NA | NA | 1 | NA |
| Studies comparable for smoke exposure | 2.6 [1.9–3.5] | [1.3–5.1] | 9 | 580 | 1189 | 1.2 [1–1.8] | 34.4 [0–69.8] | 0.142 | 0.246 |
| Studies comparable for age at interview (years) | 7.8 [3.5–17.3] | [0–1362.5] | 3 | 90 | 95 | 1.3 [1–2.2] | 36.6 [0–79.8] | 0.207 | 0.671 |
| Studies comparable for age at recrutment (months) | 4.5 [2.4–8.4] | [0.1–256.7] | 3 | 142 | 111 | 1 [1–1.7] | 0 [0–64.9] | 0.744 | 0.291 |
| Studies comparable for birth weight (grams) | 5.2 [2.2–12.2] | [0–1284.3] | 3 | 107 | 76 | 1.4 [1–2.7] | 51 [0–85.8] | 0.13 | 0.333 |
| Studies comparable for breastfeeding period (months) | 3.3 [1.9–5.6] | NA | 2 | 94 | 186 | 1 | 0 | 0.812 | NA |
| Studies comparable for height at interview (cm) | 3.4 [2.5–4.6] | [2.3–5] | 7 | 424 | 1015 | 1.2 [1–1.8] | 27.1 [0–68.4] | 0.221 | 0.328 |
| Studies comparable for number of siblings | 3 [2–4.5] | [1.2–7.2] | 4 | 188 | 372 | 1 [1–1.2] | 0 [0–34.1] | 0.874 | 0.961 |
| Studies comparable for weight at interview (kg) | 3.2 [2.3–4.3] | [2–4.9] | 6 | 378 | 923 | 1.1 [1–2.2] | 16.4 [0–78.8] | 0.308 | 0.492 |

CI: Confidence interval; OR: Odds ratio; NA: Not applicable.

Although represented by a single study in this review, pneumonia was not associated with a risk of subsequent wheezing. A previous meta-analysis showed that childhood pneumonia mainly linked to adenoviruses was associated with respiratory sequelae in hospitalized and non-hospitalized children, including restrictive pulmonary disease, obstructive pulmonary disease, bronchiectasis and chronic bronchitis [72]. Asthma, considered by most authors as ≥ 3 episodes of wheezing, were however not associated with pneumonia in the meta-analysis [72]. Although the type of LRTI was a significant source of heterogeneity during this work, we are unable to draw a definitive conclusion about this parameter because the pneumonia category was represented by a single study and the category LRTI not specified is very heterogeneous to allow an objective interpretation. Previous reports have shown that the strength of the association between LRTI in childhood and wheezing later diminished with age with persistence into

adulthood [18,70,71,73–77]. These previous reports suggest that the factor responsible for wheezing loses its influence with age, which could be small airways. Unfortunately, in this report, we have only recorded 2 studies with patients > 10 and are therefore unable to comment on the decrease in the strength of the association between LRTI in childhood and the subsequent wheezing risk.

The development of molecular diagnostic tests has made it possible to reassess the role of respiratory viruses in bronchiolitis and the description of new viruses [11]. Systematic reviews have shown that infections with no-HRSV respiratory viruses in childhood are also associated with an increased risk of developing wheezing sequelae [18,71]. Liu et al. showed the involvement of RV infections during the first 3 years of life in the subsequent wheezing in a meta-analysis including cohorts with and without controls [18]. The results of this systematic review, also, indicate that viral LRTI generally was associated with a risk of subsequent wheezing.

Studies have contributed to show that the index children with a family history of asthma had a greater risk of developing post-bronchiolitis wheezing [5]. Kneyber et al. in a quantitative analysis showed that family history of atopy/asthma were similar between index cases and controls [70]. In this systematic review, studies with confounding factor comparable in index cases and controls showed concordant results in the incidence of subsequent wheezing (Table 1). Studies have shown that male sex is a predictive factor of wheezing as a result of bronchiolitis [78,79]. In this systematic review, in the 15 studies with gender comparable between cases and controls, there were no difference in the association between early LRTI and subsequent wheezing. Studies have shown that maternal smoking was a predictor of subsequent wheezing [80]. In this meta-analysis, 4 included studies comparable for maternal smoking exposure in index cases compared to controls did not appear to affect the global effect observed. In this meta-analysis also studies comparable for the family atopy, family asthma, preponderance of overcrowding, daycare attendance, premature birth or breastfeeding did not affect the incidence of wheezing after bronchiolitis. Several plausible alternative hypotheses may be involved in post-bronchiolitis wheezing [31,81,82]. The decrease in pulmonary function very early in life has also been reported by several authors as a predisposing factor for bronchiolitis and post-bronchiolitis wheezing [35,83–85]. Despite these multiple other factors, bronchiolitis has always been shown to be a very important predictor of subsequent wheezing.

The main limitation of this systematic review is the small number of studies in some categories of our subgroup analyses. These categories included non-bronchiolitis LRTI and non-HRSV LRTI. The results of this subgroup analyses should therefore be considered with caution. The statistical significance of the symmetric or asymmetric distribution of confounders between cases and controls is largely a function of sample size. Although we used both parametric and nonparametric tests, studies with large samples have the power to detect much smaller differences between cases and controls than small studies do. We were not able to evaluate the influence of premorbid pulmonary function, the presence of co-morbidities or co-infections on the development of subsequent wheezing, as this information was missing in the majority of studies included. Despite these weaknesses, this systematic review has several strengths that include overall results consistent with multiple sensitivity analyses (consideration of the first episode of LRTI, physician-diagnosed wheezing and studies comparable for various confounding factors). No publication bias was recorded in global analyses. We conducted a comprehensive search strategy with no restriction in geographic, temporal, LRTI type, and LRTI virus type that significantly increased the number of studies and increased the statistical power of the analyses. We have adopted a transparent protocol, pre-recorded and intervention of two investigators at every stage of the process. We conducted a thorough data collection and provided an exhaustive characteristic of individual included studies. We have assessed for the first time studies comparable for confounding factors for the determination of

the association between neonatal viral LRTI and subsequent wheezing. We have transparently documented our method of defining the categories of multiple wheezing case definitions and confounding factors recorded in the included studies and the definitions considered in each category for this systematic review.

Evidence from this meta-analysis strongly suggests that viral LRTI at $\leq$ 3 years is one of the major predictors of wheezing. Despite the multiple studies conducted on the subject, the interaction between bronchiolitis and other predictors of subsequent wheezing is unknown. More controlled prospective studies are needed to resolve these confusions. Nearly all studies evaluating the long-term sequelae of bronchiolitis have focused primarily on hospitalized children [78,86,87]. One study demonstrated the involvement of mild infections induced by RV early life in the development of wheezing sequelae later [88]. It is important to implement other cohort studies to clarify the role of ambulatory bronchiolitis in developing long-term wheezing symptoms. It is also necessary to conduct additional longitudinal studies to further evaluate the role of non-HRSV respiratory viruses, especially RV and HMPV, in post-bronchiolitis wheezing. The above additional studies are expected especially from lower-middle income countries where data are very scarce.

## Supporting information

**S1 Fig. Funnel plot for publication for wheezing in children with and without LRTI in infancy.**
(PDF)

**S1 Table. PRISMA 2009 checklist.**
(PDF)

**S2 Table. Search strategy in Medline (PubMed).**
(PDF)

**S3 Table. Items for risk of bias assessment.**
(PDF)

**S4 Table. Main reasons of exclusion of eligible studies.**
(PDF)

**S5 Table. Baseline characteristics of studies meeting inclusion criteria.**
(PDF)

**S6 Table. Risk of bias assessment.**
(PDF)

**S7 Table. Individual characteristics of included studies.**
(PDF)

**S8 Table. P-value of Khi-2 and Fisher exact tests for qualitative confounding factors.**
(PDF)

**S9 Table. P-value of student test for quantitative confounding factors.**
(PDF)

**S10 Table. Subgroup analyses of wheezing in children with LRTI in infancy and control without respiratory diseases.**
(PDF)

## Author Contributions

**Conceptualization:** Sebastien Kenmoe, Richard Njouom.

**Data curation:** Sebastien Kenmoe, Arnol Bowo-Ngandji, Cyprien Kengne-Nde, Jean Thierry Ebogo-Belobo, Donatien Serge Mbaga, Gadji Mahamat, Cynthia Paola Demeni Emoh.

**Formal analysis:** Sebastien Kenmoe, Cyprien Kengne-Nde.

**Methodology:** Sebastien Kenmoe, Arnol Bowo-Ngandji, Cyprien Kengne-Nde, Jean Thierry Ebogo-Belobo, Donatien Serge Mbaga, Gadji Mahamat, Cynthia Paola Demeni Emoh, Richard Njouom.

**Project administration:** Sebastien Kenmoe, Richard Njouom.

**Supervision:** Sebastien Kenmoe, Richard Njouom.

**Validation:** Sebastien Kenmoe, Arnol Bowo-Ngandji, Cyprien Kengne-Nde, Jean Thierry Ebogo-Belobo, Donatien Serge Mbaga, Gadji Mahamat, Cynthia Paola Demeni Emoh, Richard Njouom.

**Writing – original draft:** Sebastien Kenmoe.

**Writing – review & editing:** Sebastien Kenmoe, Arnol Bowo-Ngandji, Cyprien Kengne-Nde, Jean Thierry Ebogo-Belobo, Donatien Serge Mbaga, Gadji Mahamat, Cynthia Paola Demeni Emoh, Richard Njouom.

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
