## [Decision Letter · Decision Letter 0]

29 Dec 2020

PONE-D-20-36324

Influence of confounding factors on the association between early viral LRTI and subsequent wheezing development, a systematic review and meta-analysis

PLOS ONE

Dear Dr. Njouom,

Thank you for submitting your manuscript to PLOS ONE. After careful consideration, we feel that it has merit but does not fully meet PLOS ONE’s publication criteria as it currently stands. Therefore, we invite you to submit a revised version of the manuscript that addresses the points raised during the review process.

One of the reviewers has a number of major comments and has provided an extensive review rapport to improve the manuscript. Therefore, it is important that a revised version of the manuscript has all these issues addressed and contains all improvements as suggested by this reviewer. 

We look forward to receiving your revised manuscript.

Kind regards,

Bernadette van den Hoogen

Academic Editor

PLOS ONE

Journal Requirements:

Reviewers' comments:

Reviewer's Responses to Questions

**Comments to the Author**

1. Is the manuscript technically sound, and do the data support the conclusions?

Reviewer #1: Yes

Reviewer #2: No

2. Has the statistical analysis been performed appropriately and rigorously? 

Reviewer #1: Yes

Reviewer #2: No

3. Have the authors made all data underlying the findings in their manuscript fully available?

Reviewer #1: Yes

Reviewer #2: Yes

4. Is the manuscript presented in an intelligible fashion and written in standard English?

Reviewer #1: Yes

Reviewer #2: No

5. Review Comments to the Author

Reviewer #1: This is a very-well written manuscript that proposes an update in the available results about LRTI and the risk of subsequent asthma

I would highlight in the text the mean age of evaluation, and also stress this point in the discussion (what is expected in term of respiratory prognosis in adulthood for these children?)

Reviewer #2: Thank you for the opportunity tor review this manuscript describing a meta-analytic review of studies evaluating the association between infant LRTI and subsequent wheezing illness. The study had a number of strengths, including a broad literature search, rating of study quality, consideration of a broad range of potential confounders, and duplicate coding of studies for inclusion. However, as described in detail below, I did not find the procedure for evaluating the influence of potential confounders compelling. Additionally, I found the descriptions for some critical aspects of the study procedures (e.g., operational definitions, statistical methods) to be unclear or incomplete. As a reader, I had to work hard to understand critical study procedures. For example, understanding how the authors defined “symmetrically” vs. “asymmetrically” distributed confounders required careful reading of the supplementary materials. This critical part of the study procedures should not be relegated to supplements that most readers won’t read. Finally, I do not agree with some of the authors’ interpretations of the study findings. I hope that these comments are helpful.

1. Abstract: “Our result supports absence of the influence of the main confounding factors” Frequentist null hypothesis testing can only provide evidence against the null hypothesis or inconclusive evidence. Lack of evidence for an effect is not evidence of absence. Therefore, I would recommend reworking this statement to say that the analyses did not provide evidence against the null hypothesis and avoid saying that there was evidence against the influence of the confounding factors.

2. Abstract: As PLOS One is a general scientific journal and not specifically focused on infectious airway disease, I think it’s important to define the acronyms HSRV and HMPV in the abstract rather than assuming that readers know what these acronyms mean.

3. lines 109-110: “We include only cohort studies that compare children with LRTI with controls in regards of wheezing as long-term respiratory sequelae.” I think it would be helpful to reader to have an explanation of why other studies (e.g., case-control studies) were excluded. Cohort studies have advantages over case-control studies that could potentially justify this decision, but I think it’s important to explain the rationale clearly.

4. lines 115-117: “Studies with only high-risk participants (preterm, chronic respiratory disease, heart disease or immunodeficiency) were not considered.” Again, why were these studies excluded? Is it not of interest whether LRTIs have a causal effect on chronic respiratory illness in these high-risk populations?

5. lines 122-123: “Wheezing was considered as any whistling nose at the expiration declared by the included study.” This doesn’t seem like a clear enough operationalization of the study outcome. Did you include any wheezing outcomes, including parental report?

6. Did the authors require a washout period between the LRTI and measurement of the wheezing outcomes to ensure that the wheezing was due to a subsequent illness and was not merely a symptom of the exposure-defining LRTI? That is, can we be fairly certain that the exposure and outcome variables were due to separate illnesses?

7. line 123: “Wheezing during follow up was defined as current wheezing.” This again is not clear to me. Does current wheezing mean that the child was wheezing during the evaluation or assessment?

8. I don’t think the statistical analysis section had sufficient detail. Did the authors use a fixed-effects models? Random effects models? If the authors used random-effects meta-analysis (this seems more appropriate than fixed effects), what method was used to estimate the between-study variable (e.g., DerSimonian Laird, Paule-Mandel, Hartung-Knapp)? How did the authors deal with the fact that many studies report more than one relevant estimate? For example, some studies report effect estimates for the same construct (e.g., current wheeze) at multiple time points or using different measures of the same construct. Did they average across estimates? Did they select specific estimates to include and, if so, how was it determined which estimates to include? Did the authors calculate naïve odds ratios or did they use adjusted odds ratios when they were available?

9. lines 156-157: “The validity of results was evaluated by a sensitivity analysis including only first episode of LRTI and studies with confounding factors distributed symmetrically between cases and controls.” How are the authors defining “symmetrically distributed”? What is the criterion used to determine whether confounders were symmetrically distributed? I gather from the S7 Table that the authors conducted chi-square tests evaluating whether the confounding factors were independent of LRTI exposure and used the p value to determine “symmetrically” vs. “asymmetrically” distributed confounding factors. This approach should be described clearly in the Method section. If this is the method that what used, I don’t think it is a compelling approach. Statistical significance is largely a function of sample size. Studies with large samples have the power to detect much smaller discrepancies between LRTI+ and LRTI- groups than small studies do. So, a small study could have an equal or larger discrepancy on a confounder between the LRTI+ and LRTI- groups than a much larger study, yet be labeled as “symmetric” simply because it had less power to detect the discrepancy. Relatedly, the fact that there is no significant difference between the exposed and non-exposed groups on levels of a confounder does not mean that the confounder cannot bias the estimate of the effect of the exposure on the outcome. Again, with small sample sizes, there could be important imbalances that are not statistically significant. In sum, I am not convinced that the variable coding studies as having “asymmetrically” vs. “symmetrically” distributed confounders provides a reasonable approximation for the level of balance on the confounders. Therefore, I am not sure if the sensitivity analysis including only symmetrical studies has great value.

10. Table 1 is not adequately labeled or described. Do the rows represent subgroup analyses? For example, does the row labeled “Heredity for asthma” provide an estimate for the subgroup of estimates from samples that were positive for asthma heredity? Or does this row provide an aggregate estimate among the subgroup of studies for which there was no significant difference between the LRTI+ and LRTI- groups on heredity for asthma? How was heredity for asthma defined?

11. line 160: “We analyzed the data using the R software version 3.5.1.” It would be good to also provide the version of the meta package used to conduct the analyses.

12. line 181: “All included studies were at low risk of bias.” This statement seems too vague. In the supplement, the authors note that a score of 6-9 on the Newcastle-Ottawa scale was indicative of low bias. Were there any areas of concern across the studies? For example, did all of the studies have low risk of bias due to missingness? This seems unlikely.

13. In Table S9, the authors provide the “P-value subgroup difference” in the final column. But there I did not see any information about how this p-value was obtained. Did the authors run multiple meta-analytic models with these study characteristics entered as covariates to obtain the p-values? Was one covariate entered per model or were multiple covariates entered in the same model? Generally speaking, I think more information is needed about the modeling approach. Why did the authors include 95% prediction intervals in the table? How should the readers interpret this and why is it needed in addition to the 95% confidence interval?

14. lines 212-214: “The type of LRTI was associated with the occurrence of wheezing (p= 0.007) (S9 Table). History of bronchiolitis (OR= 3.8, 95% CI= 3.4-4.7) and LRTI unspecified (OR= 1.9, 95% CI= 1.4-2.7) were associated with the occurrence of wheezing in childhood.” If I understand correctly, the type of LRTI accounted for significant heterogeneity in the effect estimates. This means that the strength of the estimated effect of LRTI on wheezing outcomes differed depending on the type of LRTI. But the authors don’t explain which comparison was driving this effect. They note that studies using both bronchiolitis and unspecified LRTI found significant weighted mean effect sizes. The third category (studies evaluating pneumonia as the exposure) did not yield a significant effect size, though this is likely due to the fact that there was only one pneumonia exposure study. The point estimate for the pneumonia study (OR=2.8) was larger than the point estimate and the upper bound of the 95% CI for the unspecified bronchiolitis studies. So, we shouldn’t make much of the fact that there was too much uncertainty in the pneumonia study to reject the null hypothesis. My guess is that the weighted mean OR for the bronchiolitis studies was significantly larger than the weighted mean OR for the unspecified studies and this is why LRTI type accounts for significant heterogeneity. The 95% Cis for these two study groups (bronchiolitis and unspecified) do not overlap at all. But the meaning of the significant effect of LRTI type does not come across in the authors’ description at all.

15. lines 216-218: “The type of LRTI was associated with the occurrence of wheezing (p= 0.007) (S9 Table). History of bronchiolitis (OR= 3.8, 95% CI= 3.4-4.7) and LRTI unspecified (OR= 1.9, 95% CI= 1.4-2.7) were associated with the occurrence of wheezing in childhood.” The authors present what appears to be an analysis of whether “Age range at recruitment” accounts for significant between-study heterogeneity in effect estimates. Again, however, the authors follow up a significant effect of this covariate with an analysis of which subgroups (levels of the covariate) yielded significant weighted mean ORs. There is no focus on which levels of the covariate differ from each other (e.g., Is there a significant difference in the mean OR for the studies beginning recruitment before 6 months compared to studies beginning recruitment before 1 year of age?), making the effect of the covariate hard to interpret for readers. Furthermore, the authors created a categorical “Age at recruitment” variable with four levels. However, two cells of this variable are highly sparse, containing one and two studies. I don’t think the authors have enough data to support a four-level factor for this analysis.

16. lines 218-219: “We did not record any difference in the occurrence of wheezing by type of virus screened (p= 0.249).” I don’t think that this analysis was very meaningful as 20 of the 22 studies evaluated HRSV. There is simply not enough information to conduct a meaningful analysis.

17. lines 221-222: “No difference was recorded in the occurrence of wheezing depending on the sampling approach (non-probabilistic vs probabilistic, p= 0.505)” Again, I don’t think that this was a meaningful analysis as there was only one study that used probabilistic sampling.

18. lines 228-229: “Children who experienced an episode of HMPV or HRSV bronchiolitis at ≤ 3 years of age were 3 times more likely than controls to develop wheezing later.” It’s unclear to me why the main conclusion is limited to HMPV and HRSV bronchiolitis rather than LRTI more broadly. The main hypothesis was about LRTI more generally. Why are we drawing conclusions about specific subgroups rather than LRTI altogether?

19. lines 229-232: “This increased risk of wheezing in children with a history of HMPV or HRSV bronchiolitis was not influenced by the rank of bronchiolitis episode and any of the investigated confounding factors.” It’s not clear to me what “rank of bronchiolitis episode” means. Additionally, I don’t think that the authors truly tested whether confounding factors influenced the estimates. As noted in an earlier comment, I don’t think the procedure for determining “symmetrically” and “asymmetrically” distributed confounders was compelling. It would have been far more interesting to evaluate whether estimates based on adjustment for specific covariates believed to represent major confounders (e.g., genetic confounding) were smaller than estimates that did not adjust for these factors.

20. lines 248-249: “Asthma, considered by most authors as ≥ 3 episodes of wheezing, were however not associated with pneumonia in the meta-analysis.” This comment must be put in context. According to Table S9, there was only one pneumonia study with 40 participants. This meta-analysis does not provide us any new information beyond what the one pneumonia study has already provided because it simply reports the results of this one trial – there is not combining of information across pneumonia studies.

21. lines 255-257: “The results of this systematic review, also, indicate that apart of HRSV, HMPV bronchiolitis was associated with a risk of subsequent wheezing.” Again, according to Table S9, there was only one HMPV study. There was no need for meta-analytic aggregation because there was only a single estimate, from what I can tell. In which case, this review did not provide any new information about the association between HMPV and subsequent wheezing above what was already reported from the single study that evaluated the association between HMPV LRTI and wheezing illness. There is no unique contribution of the meta-analysis in relation to the estimated effect of HMPV LRTI on subsequent wheezing illness.

22. line 261: “In this systematic review, studies with confounding factor similarly distributed” This is an important point, so I will make it again. The fact that there was no statistically significant imbalance across the LRTI+ and LRTI- groups does not necessarily mean that the confounding variable was similarly distributed across the two groups.

6. PLOS authors have the option to publish the peer review history of their article (what does this mean?). If published, this will include your full peer review and any attached files.

Reviewer #1: No

Reviewer #2: No

---

## [Author Response · Author response to Decision Letter 0]

6 Jan 2021

Review Comments to the Author

Reviewer #1: This is a very-well written manuscript that proposes an update in the available results about LRTI and the risk of subsequent asthma

Authors: Thank you for appreciation and summary.

I would highlight in the text the mean age of evaluation, and also stress this point in the discussion (what is expected in term of respiratory prognosis in adulthood for these children?)

Authors: Thank you for this suggestion. We added to the manuscript a subgroup analysis by age of participants at interview in the results and discussion sections. See below corresponding text:

“Wheezing in children was reported between 2 and 20 years and most studies were between 2 and 10 years (77.2%).

No difference was recorded in the occurrence of wheezing depending on the age range at the interview (p= 0.053)

However, all age groups at interview up to 20 years were associated with an increased risk of developing wheezing in former cases of LRTI compared to controls.

Previous reports have shown that the strength of the association between LRTI in childhood and wheezing later diminished with age with persistence into adulthood [18,67,68,70–74]. These previous reports suggest that the factor responsible for wheezing loses its influence with age, which could be small airways. Unfortunately, in this report, we have only recorded 2 studies with patients > 10 and are therefore unable to comment on the decrease in the strength of the association between LRTI in childhood and the subsequent wheezing risk.”

Reviewer #2: Thank you for the opportunity tor review this manuscript describing a meta-analytic review of studies evaluating the association between infant LRTI and subsequent wheezing illness. The study had a number of strengths, including a broad literature search, rating of study quality, consideration of a broad range of potential confounders, and duplicate coding of studies for inclusion. However, as described in detail below, I did not find the procedure for evaluating the influence of potential confounders compelling. Additionally, I found the descriptions for some critical aspects of the study procedures (e.g., operational definitions, statistical methods) to be unclear or incomplete. As a reader, I had to work hard to understand critical study procedures. For example, understanding how the authors defined “symmetrically” vs. “asymmetrically” distributed confounders required careful reading of the supplementary materials. This critical part of the study procedures should not be relegated to supplements that most readers won’t read. Finally, I do not agree with some of the authors’ interpretations of the study findings. I hope that these comments are helpful.

Authors: Thank you for this summary.

1. Abstract: “Our result supports absence of the influence of the main confounding factors” Frequentist null hypothesis testing can only provide evidence against the null hypothesis or inconclusive evidence. Lack of evidence for an effect is not evidence of absence. Therefore, I would recommend reworking this statement to say that the analyses did not provide evidence against the null hypothesis and avoid saying that there was evidence against the influence of the confounding factors.

Authors: Thank you for these suggestions. The abstract has been modified and can now be read as below: 

“The risk of developing subsequent wheezing was conserved when considering studies with comparable groups for socio-demographic and clinical confounders.

When considering studies with comparable groups for most confounding factors, our results provided strong evidence for the association between neonatal HRSV or HMPV bronchiolitis and the subsequent wheezing development.”

2. Abstract: As PLOS One is a general scientific journal and not specifically focused on infectious airway disease, I think it’s important to define the acronyms HSRV and HMPV in the abstract rather than assuming that readers know what these acronyms mean.

Authors: Thank you, the text has been corrected as suggested

3. lines 109-110: “We include only cohort studies that compare children with LRTI with controls in regards of wheezing as long-term respiratory sequelae.” I think it would be helpful to reader to have an explanation of why other studies (e.g., case-control studies) were excluded. Cohort studies have advantages over case-control studies that could potentially justify this decision, but I think it’s important to explain the rationale clearly.

Authors: Thank you for this suggestion. The text has been modified and can now be read as below: 

“Due to the multiple limitations of cross-sectional (non-comparable groups) and case control (misclassification and recall bias) studies in estimating the association between exposure and outcome, we include only cohort studies that compare children with LRTI with controls in regards of wheezing as long-term respiratory sequelae.”

4. lines 115-117: “Studies with only high-risk participants (preterm, chronic respiratory disease, heart disease or immunodeficiency) were not considered.” Again, why were these studies excluded? Is it not of interest whether LRTIs have a causal effect on chronic respiratory illness in these high-risk populations?

Authors: We thank the reviewer for this excellent comment. The long-term sequelae of lower respiratory infections in childhood in subjects with underlying medical conditions have been the subject of numerous observational and interventional studies and are of considerable interest. This aspect is not, however, part of the objectives of this review. Given the critical importance of this research question, we have already recorded a second protocol at Prospero (CRD42019131343) to more specifically address the concern in another ongoing review.

5. lines 122-123: “Wheezing was considered as any whistling nose at the expiration declared by the included study.” This doesn’t seem like a clear enough operationalization of the study outcome. Did you include any wheezing outcomes, including parental report?

Authors: Thank you for this excellent comment. We included all wheezing including those defined by parents. We added all wheezing definitions from included studies in S6 Table. We also conducted a sensitivity analysis including only studies with physician-diagnosed wheezing, and it does not reflect any major differences from overall results.

6. Did the authors require a washout period between the LRTI and measurement of the wheezing outcomes to ensure that the wheezing was due to a subsequent illness and was not merely a symptom of the exposure-defining LRTI? That is, can we be fairly certain that the exposure and outcome variables were due to separate illnesses?

Authors: As shown in S6 Table, the minimum time between the onset of the lower respiratory infection and the development of wheezing was 1.5 years. We also considered from the inclusion criteria predefined in the protocol declared in Prospero to include only studies with a minimum follow-up duration of one year, which suggests a sufficiently long duration for wheezing (outcome) recorded to not be a symptom of lower respiratory disease (exposure). Thank you.

7. line 123: “Wheezing during follow up was defined as current wheezing.” This again is not clear to me. Does current wheezing mean that the child was wheezing during the evaluation or assessment?

Authors: Thank you for this suggestion. The text has been modified and can now be read as below:

“Wheezing during the interview was defined as current wheezing. Wheezing in the last 12 months was defined as wheezing episodes during the previous year. Recurrent wheezing was considered for persistent wheezing between the diagnosis of LRTI and the interview.”

8. I don’t think the statistical analysis section had sufficient detail. Did the authors use a fixed-effects models? Random effects models? If the authors used random-effects meta-analysis (this seems more appropriate than fixed effects), what method was used to estimate the between-study variable (e.g., DerSimonian Laird, Paule-Mandel, Hartung-Knapp)? How did the authors deal with the fact that many studies report more than one relevant estimate? For example, some studies report effect estimates for the same construct (e.g., current wheeze) at multiple time points or using different measures of the same construct. Did they average across estimates? Did they select specific estimates to include and, if so, how was it determined which estimates to include? Did the authors calculate naïve odds ratios or did they use adjusted odds ratios when they were available?

Authors: Thank you for this suggestion. The text has been modified and can now be read as below:

“If there were multiple wheezing phenotypes in the same study, we chose the one that spanned the longest time. We separately considered studies with multiple effect data depending on interview time, respiratory viruses detected or types of LRTI.

The odds ratio (OR) and the 95% confidence interval (95% CI) were used as a measure of the association between LRTI and long-term wheezing. These parameters were calculated using a random-effects model by the Der Simonian and Laird method [45]. Heterogeneity was assessed by the Q test p-value and the I² statistic [46,47]. The heterogeneity between the studies was considered significant for the values of P <0.1 and I²> 50%. The robustness of the results was assessed by a sensitivity analysis including only the first episode of LRTI, physician-diagnosed wheezing and studies with proportions of confounding factors comparable between cases and controls. The comparability of confounders between cases and controls were estimated using the Chi-square and Fisher tests for qualitative variables and Student's T-test for continuous variables. The primary confounding factors data for continuous variables expressed as median and/or range were converted to mean and standard deviation [48]. Subgroup analyses were performed based on sampling method, time of exposure collection, age range during LRTI, age at time of interview for wheezing, hospitalization of controls, viruses responsible for LRTI, type of LRTI, and type of wheezing. We analysed the data using version 3.5.1 of software R [49,50].”

9. lines 156-157: “The validity of results was evaluated by a sensitivity analysis including only first episode of LRTI and studies with confounding factors distributed symmetrically between cases and controls.” How are the authors defining “symmetrically distributed”? What is the criterion used to determine whether confounders were symmetrically distributed? I gather from the S7 Table that the authors conducted chi-square tests evaluating whether the confounding factors were independent of LRTI exposure and used the p value to determine “symmetrically” vs. “asymmetrically” distributed confounding factors. This approach should be described clearly in the Method section. 

Authors: Thank you for this comment. We have now described in full detail our definition of “symmetric distribution” and the methodological approach we used to obtain the results. The text has been modified and can now be read as below:

“The robustness of the results was assessed by a sensitivity analysis including only the first episode of LRTI, physician-diagnosed wheezing and studies with proportions of confounding factors comparable between cases and controls. The comparability of confounders between cases and controls were estimated using the Chi-square and Fisher tests for qualitative variables and Student's T-test for continuous variables. The primary confounding factors data for continuous variables expressed as median and/or range were converted to mean and standard deviation [48].”

If this is the method that what used, I don’t think it is a compelling approach. Statistical significance is largely a function of sample size. Studies with large samples have the power to detect much smaller discrepancies between LRTI+ and LRTI- groups than small studies do. So, a small study could have an equal or larger discrepancy on a confounder between the LRTI+ and LRTI- groups than a much larger study, yet be labeled as “symmetric” simply because it had less power to detect the discrepancy. Relatedly, the fact that there is no significant difference between the exposed and non-exposed groups on levels of a confounder does not mean that the confounder cannot bias the estimate of the effect of the exposure on the outcome. Again, with small sample sizes, there could be important imbalances that are not statistically significant. In sum, I am not convinced that the variable coding studies as having “asymmetrically” vs. “symmetrically” distributed confounders provides a reasonable approximation for the level of balance on the confounders. Therefore, I am not sure if the sensitivity analysis including only symmetrical studies has great value.

Authors: We greatly thank the reviewer for these clarifications. We used a parametric test (chi-square test) and a non-parametric test (fisher test) to try to attenuate the effect of this small population size on the statistical significance of the distribution of confusing factors between cases and controls. We have now pointed out this aspect in the limits section of our discussion. We fully agree with the Reviewer that a symmetrical allocation of the confounding factor between cases and controls does not guarantee the exclusion of its influence on the effect of exposure on outcome. However, it is known that comparable samples on the majority of confounding factors give a result with a higher level of proof than those which are not. In this regard, the randomized trials which are known to have comparable sample according to most of the confounding factors. Studies paired for certain sociodemographic confounding factors such as sex and age are also accepted as comparable for these aspects. However, the examination of the comparability of the remaining confounders is most often done on the basis of the statistical significance of the tests performed, as we have done in the present work.

10. Table 1 is not adequately labeled or described. Do the rows represent subgroup analyses? For example, does the row labeled “Heredity for asthma” provide an estimate for the subgroup of estimates from samples that were positive for asthma heredity? Or does this row provide an aggregate estimate among the subgroup of studies for which there was no significant difference between the LRTI+ and LRTI- groups on heredity for asthma? How was heredity for asthma defined?

Authors: Thank you for these comments. We have now improved the label of row entities. All confounding factors presented have been defined as presented in the included article. We have harmonized the names of the confounding factors collected in several articles according to their similarity. We have now clarified this in the section “data extraction”.

11. line 160: “We analyzed the data using the R software version 3.5.1.” It would be good to also provide the version of the meta package used to conduct the analyses.

Authors: Thank you for the suggestion the packages used are now specified.

12. line 181: “All included studies were at low risk of bias.” This statement seems too vague. In the supplement, the authors note that a score of 6-9 on the Newcastle-Ottawa scale was indicative of low bias. Were there any areas of concern across the studies? For example, did all of the studies have low risk of bias due to missingness? This seems unlikely.

Authors: We added S6 Table which presents the individual score allocations for the included studies and the total scores to rule out this ambiguity. Thank you.

13. In Table S9, the authors provide the “P-value subgroup difference” in the final column. But there I did not see any information about how this p-value was obtained. Did the authors run multiple meta-analytic models with these study characteristics entered as covariates to obtain the p-values? Was one covariate entered per model or were multiple covariates entered in the same model? Generally speaking, I think more information is needed about the modeling approach. Why did the authors include 95% prediction intervals in the table? How should the readers interpret this and why is it needed in addition to the 95% confidence interval?

Authors: Thank you for the feedback, we performed the subgroup analyses using random-effect meta-analysis with single-covariate analyses. Although the interpretation of the prediction interval can be difficult, this parameter provides a convenient format for expressing any uncertainty surrounding the effect size because it takes into account the magnitude and consistency of the effects. The prediction interval thus represents one of the most important results of a meta-analysis because the aim of the work is to put the acquired knowledge into future application. We have now added the relevant references in the manuscript and modify the text accordingly.

14. lines 212-214: “The type of LRTI was associated with the occurrence of wheezing (p= 0.007) (S9 Table). History of bronchiolitis (OR= 3.8, 95% CI= 3.4-4.7) and LRTI unspecified (OR= 1.9, 95% CI= 1.4-2.7) were associated with the occurrence of wheezing in childhood.” If I understand correctly, the type of LRTI accounted for significant heterogeneity in the effect estimates. This means that the strength of the estimated effect of LRTI on wheezing outcomes differed depending on the type of LRTI. But the authors don’t explain which comparison was driving this effect. They note that studies using both bronchiolitis and unspecified LRTI found significant weighted mean effect sizes. The third category (studies evaluating pneumonia as the exposure) did not yield a significant effect size, though this is likely due to the fact that there was only one pneumonia exposure study. The point estimate for the pneumonia study (OR=2.8) was larger than the point estimate and the upper bound of the 95% CI for the unspecified bronchiolitis studies. So, we shouldn’t make much of the fact that there was too much uncertainty in the pneumonia study to reject the null hypothesis. My guess is that the weighted mean OR for the bronchiolitis studies was significantly larger than the weighted mean OR for the unspecified studies and this is why LRTI type accounts for significant heterogeneity. The 95% Cis for these two study groups (bronchiolitis and unspecified) do not overlap at all. But the meaning of the significant effect of LRTI type does not come across in the authors’ description at all.

Authors: Thank you for these relevant comments. We added the following sentence in the discussion section:

“Although the type of LRTI was a significant source of heterogeneity during this work, we are unable to draw a definitive conclusion about this parameter because the pneumonia category was represented by a single study and the category LRTI not specified is very heterogeneous to allow an objective interpretation.”

15. lines 216-218: “The type of LRTI was associated with the occurrence of wheezing (p= 0.007) (S9 Table). History of bronchiolitis (OR= 3.8, 95% CI= 3.4-4.7) and LRTI unspecified (OR= 1.9, 95% CI= 1.4-2.7) were associated with the occurrence of wheezing in childhood.” The authors present what appears to be an analysis of whether “Age range at recruitment” accounts for significant between-study heterogeneity in effect estimates. Again, however, the authors follow up a significant effect of this covariate with an analysis of which subgroups (levels of the covariate) yielded significant weighted mean ORs. There is no focus on which levels of the covariate differ from each other (e.g., Is there a significant difference in the mean OR for the studies beginning recruitment before 6 months compared to studies beginning recruitment before 1 year of age?), making the effect of the covariate hard to interpret for readers. Furthermore, the authors created a categorical “Age at recruitment” variable with four levels. However, two cells of this variable are highly sparse, containing one and two studies. I don’t think the authors have enough data to support a four-level factor for this analysis.

Authors: We totally agree with the Reviewer, thanks for the comment. We combined categories <2 and <3, consisting of only 3 studies, and redone the analyses. We have modified the result section accordingly. Thank you.

16. lines 218-219: “We did not record any difference in the occurrence of wheezing by type of virus screened (p= 0.249).” I don’t think that this analysis was very meaningful as 20 of the 22 studies evaluated HRSV. There is simply not enough information to conduct a meaningful analysis.

Authors: We initially pointed out this major limitation of our work in the discussion (see sentence below). We have also withdrawn the conclusions of our work drawn from this subgroup analysis and defined the corresponding research implications in the discussion section. Thank you. 

“The main limitation of this systematic review is the small number of studies in some categories of our subgroup analyses. These categories included non-bronchiolitis LRTI and non-HRSV LRTI. The results of this subgroup analyses should therefore be considered with caution.”

17. lines 221-222: “No difference was recorded in the occurrence of wheezing depending on the sampling approach (non-probabilistic vs probabilistic, p= 0.505)” Again, I don’t think that this was a meaningful analysis as there was only one study that used probabilistic sampling.

Authors: We initially pointed out this major limitation of our work in the discussion (see sentence below). We have also withdrawn the conclusions of our work drawn from this subgroup analysis and defined the corresponding research implications in the discussion section. Thank you. 

“The main limitation of this systematic review is the small number of studies in some categories of our subgroup analyses. These categories included non-bronchiolitis LRTI and non-HRSV LRTI. The results of this subgroup analyses should therefore be considered with caution.”

18. lines 228-229: “Children who experienced an episode of HMPV or HRSV bronchiolitis at ≤ 3 years of age were 3 times more likely than controls to develop wheezing later.” It’s unclear to me why the main conclusion is limited to HMPV and HRSV bronchiolitis rather than LRTI more broadly. The main hypothesis was about LRTI more generally. Why are we drawing conclusions about specific subgroups rather than LRTI altogether?

Authors: Subgroup analyses do obviously present a major weakness due to the low number of studies in most categories. We thus consider the global null hypothesis throughout the manuscript. Thank you.

19. lines 229-232: “This increased risk of wheezing in children with a history of HMPV or HRSV bronchiolitis was not influenced by the rank of bronchiolitis episode and any of the investigated confounding factors.” It’s not clear to me what “rank of bronchiolitis episode” means. Additionally, I don’t think that the authors truly tested whether confounding factors influenced the estimates. As noted in an earlier comment, I don’t think the procedure for determining “symmetrically” and “asymmetrically” distributed confounders was compelling. It would have been far more interesting to evaluate whether estimates based on adjustment for specific covariates believed to represent major confounders (e.g., genetic confounding) were smaller than estimates that did not adjust for these factors.

Authors: We reviewed our initial misleading consideration of confounding factors throughout the manuscript. Rather, our sensitivity analysis takes into account studies that are comparable according to the confounding factors collected. In our work the effects are calculated from primary numbers collected in the included studies. We are therefore unable to collect directly adjusted versus unadjusted effects. Thank you.

20. lines 248-249: “Asthma, considered by most authors as ≥ 3 episodes of wheezing, were however not associated with pneumonia in the meta-analysis.” This comment must be put in context. According to Table S9, there was only one pneumonia study with 40 participants. This meta-analysis does not provide us any new information beyond what the one pneumonia study has already provided because it simply reports the results of this one trial – there is not combining of information across pneumonia studies.

Authors: We totally agree with the Reviewer. This sentence was the description of a result of the review cited in the previous sentence “A previous meta-analysis showed that childhood pneumonia mainly linked to adenoviruses was associated with respiratory sequelae in hospitalized and non-hospitalized children, including restrictive pulmonary disease, obstructive pulmonary disease, bronchiectasis and chronic bronchitis [72]”. We repeated the citation to rule out the confusion. Thank you.

21. lines 255-257: “The results of this systematic review, also, indicate that apart of HRSV, HMPV bronchiolitis was associated with a risk of subsequent wheezing.” Again, according to Table S9, there was only one HMPV study. There was no need for meta-analytic aggregation because there was only a single estimate, from what I can tell. In which case, this review did not provide any new information about the association between HMPV and subsequent wheezing above what was already reported from the single study that evaluated the association between HMPV LRTI and wheezing illness. There is no unique contribution of the meta-analysis in relation to the estimated effect of HMPV LRTI on subsequent wheezing illness.

Authors: Subgroup analyses do obviously present a major weakness due to the low number of studies in most categories. We thus consider the global null hypothesis throughout the manuscript. Thank you.

22. line 261: “In this systematic review, studies with confounding factor similarly distributed” This is an important point, so I will make it again. The fact that there was no statistically significant imbalance across the LRTI+ and LRTI- groups does not necessarily mean that the confounding variable was similarly distributed across the two groups.

Authors: We fully agree with the Reviewer that a symmetrical allocation of the confounding factor between cases and controls does not guarantee the exclusion of its influence on the effect of exposure on outcome. However, it is known that comparable samples on the majority of confounding factors give a result with a higher level of proof than those which are not. In this regard, the randomized trials which are known to have comparable sample according to most of the confounding factors. Studies paired for certain sociodemographic confounding factors such as sex and age are also accepted as comparable for these aspects. However, the examination of the comparability of the remaining confounders is most often done on the basis of the statistical significance of the tests performed, as we have done in the present work. Other best study designs also include adjustment of the estimated effect on major confounders. However, we calculate the effects in this work using primary data from the included studies. We do not collect the estimated effects in the study. Thank you.

---

## [Decision Letter · Decision Letter 1]

19 Mar 2021

PONE-D-20-36324R1

Association between early viral LRTI and subsequent wheezing development, a meta-analysis and sensitivity analyses for studies comparable for confounding factors.

PLOS ONE

Dear Dr. Njouom,

Thank you for submitting your manuscript to PLOS ONE. After careful consideration, we feel that it has merit but does not fully meet PLOS ONE’s publication criteria as it currently stands. Therefore, we invite you to submit a revised version of the manuscript that addresses the points raised during the review process.

We look forward to receiving your revised manuscript.

Kind regards,

Bernadette van den Hoogen

Academic Editor

PLOS ONE

Journal Requirements:

Reviewers' comments:

Reviewer's Responses to Questions

**Comments to the Author**

1. If the authors have adequately addressed your comments raised in a previous round of review and you feel that this manuscript is now acceptable for publication, you may indicate that here to bypass the “Comments to the Author” section, enter your conflict of interest statement in the “Confidential to Editor” section, and submit your "Accept" recommendation.

Reviewer #1: All comments have been addressed

Reviewer #3: All comments have been addressed

2. Is the manuscript technically sound, and do the data support the conclusions?

Reviewer #1: (No Response)

Reviewer #3: Yes

3. Has the statistical analysis been performed appropriately and rigorously? 

Reviewer #1: (No Response)

Reviewer #3: Yes

4. Have the authors made all data underlying the findings in their manuscript fully available?

Reviewer #1: (No Response)

Reviewer #3: Yes

5. Is the manuscript presented in an intelligible fashion and written in standard English?

Reviewer #1: (No Response)

Reviewer #3: Yes

6. Review Comments to the Author

Reviewer #1: (No Response)

Reviewer #3: I thank the editors for the opportunity to review this manuscript.

This manuscript is clearly written and have adequately addressed the comments from the previous reviewers. I agree with the points made by reviewer 2 and have additional minor comments stated below.

1) Please explain rationale on exclusion of high risk population in the paper

2) I am not sure why proportions of confounders similar in both cases and controls were used in sensitivity analysis. Is this to ensure that the effects seen is not due to confounding? But I will assume that the studies have corrected for confounders in their analysis?

7. PLOS authors have the option to publish the peer review history of their article (what does this mean?). If published, this will include your full peer review and any attached files.

Reviewer #1: No

Reviewer #3: No

---

## [Author Response · Author response to Decision Letter 1]

21 Mar 2021

Review Comments to the Author

Reviewer #3: I thank the editors for the opportunity to review this manuscript.

This manuscript is clearly written and have adequately addressed the comments from the previous reviewers. I agree with the points made by reviewer 2 and have additional minor comments stated below.

Authors: We thank Reviewer 1 for these favorable comments.

1) Please explain rationale on exclusion of high risk population in the paper

Authors: Studies often recruit high-risk cases (eg preterm) and not for the controls (eg apparently healthy individuals). This type of study is therefore likely to lead to a confounding bias in attributing the observed effect to exposure and yet it is rather the comorbidity in high-risk subjects that is linked to the effect. We have now clarified in the manuscript that we have excluded studies with high-risk individuals to reduce the potentially confounding effect associated with these studies. Thank you 

2) I am not sure why proportions of confounders similar in both cases and controls were used in sensitivity analysis. Is this to ensure that the effects seen is not due to confounding? But I will assume that the studies have corrected for confounders in their analysis?

Authors: We claim that studies with similar proportions of confounding factors in both cases and controls are associated with a higher level of proof for the association between exposure (LRTI) and outcome (wheezing). Thank you.

---

## [Editor Report · Decision Letter 2]

26 Mar 2021

Association between early viral LRTI and subsequent wheezing development, a meta-analysis and sensitivity analyses for studies comparable for confounding factors.

PONE-D-20-36324R2

Dear Dr. Njouom,

We’re pleased to inform you that your manuscript has been judged scientifically suitable for publication and will be formally accepted for publication once it meets all outstanding technical requirements.

Kind regards,

Bernadette van den Hoogen

Academic Editor

PLOS ONE
---

## [Editor Report · Acceptance letter]

30 Mar 2021

PONE-D-20-36324R2 

Association between early viral LRTI and subsequent wheezing development, a meta-analysis and sensitivity analyses for studies comparable for confounding factors. 

Dear Dr. Njouom:

I'm pleased to inform you that your manuscript has been deemed suitable for publication in PLOS ONE. Congratulations! Your manuscript is now with our production department. 

Kind regards, 

on behalf of

Dr. Bernadette van den Hoogen 

Academic Editor

PLOS ONE